# Clinical Presentations of Adolescents Aged 16–18 Years in the Adult Emergency Department

**DOI:** 10.3390/ijerph18189578

**Published:** 2021-09-11

**Authors:** Lara Aurora Brockhus, Martina Bärtsch, Aristomenis K. Exadaktylos, Kristina Keitel, Jolanta Klukowska-Rötzler, Martin Müller

**Affiliations:** 1Department of Emergency Medicine, Inselspital, Bern University Hospital, Bern University, 3010 Bern, Switzerland; martina.baertsch@students.unibe.ch (M.B.); aristomenis.exadaktylos@insel.ch (A.K.E.); 2Department of Paediatric Emergency Medicine, University Children’s Hospital, Inselspital, Bern University, 3010 Bern, Switzerland; kristina.keitel@insel.ch; 3Department of Paediatrics, University Children’s Hospital, Inselspital, Bern University, 3010 Bern, Switzerland

**Keywords:** ED presentation, adolescents 16–18 years, comparison adults and adolescents

## Abstract

Background: In many large hospitals in Switzerland, adolescents 16 years and older are treated in adult emergency departments (ED). There have been few publications about this specific patient population, especially in Switzerland. This study aims to provide an overview of emergency presentations of adolescents between 16–18 years of age when compared to adults and focuses on their principle complaints. Methods: We conducted a single-centre, retrospective, cross-sectional study of all patients aged 16 years and older presenting to the adult ED at the University Hospital (Inselspital) in Bern, Switzerland, from 2013 to 2017. This analysis gives an overview of emergency presentations of adolescents between 16–18 years of age in this time period and compares their consultation characteristics to those of adult patients. Results: Data of a total of 203,817 patients who presented to our adult ED between 2013 and 2017 were analysed. Adolescents account for 2.5% of all emergency presentations. The number of ED presentations in the reviewed time period rose for adults (+2368, 95% CI: 1695, 3041, *p* = 0.002 consultations more per year; +25% comparing 2013 with 2017), while adolescent presentations did not significantly increase (*p* = 0.420). In comparison to adult patients, adolescents presented significantly more often during the night (39.1% vs. 31.5%, *p* < 0.001), as walk-ins (54.2% vs. 44.9%, *p* < 0.001), or with less highly acute complaints at triage (21% vs. 31%, *p* < 0.001). They were more likely to be discharged (70.8% vs. 52.2%, *p* < 0.001). We found a significant association between the two age groups and principle complaints. In comparison to adults, trauma and psychiatric problems were significantly more common among adolescents. Conclusions: Our data showed that complaints in adolescent patients under 18 years of age significantly differ from those in older patients. The artificial age cut-off therefore puts this vulnerable population at risk of receiving inadequate diagnostic testing and treatment adapted only for adults. Additional studies are needed on the reasons adolescents and young adults seek ED care, as this could lead to improvements in the care processes for this vulnerable population.

## 1. Introduction

Consultations to the ED are rising in most Western countries [1,2,3]; in our emergency department (ED), the increase between 2013 and 2019 was 32% (38,027 cases in 2013 and 50,033 cases in 2019) [4,5].

Increasing ED use has also been observed in other studies of young adults [6,7], as there are often no appropriate primary care services for this physical, developmental, and social transition phase. Another reason for this could be that the ED may be the only contact point for adolescents [8]. The UN Convention on the Rights of a Child defines adolescents as individuals between the ages of 10 and 19 years [9]. This challenge is also present in Switzerland, as there are no dedicated primary care services for adolescents [10]. The adolescent ED population is defined and served differently from country to country [9,11]. In Switzerland, most EDs have established an artificial cut-off at 16 years and therefore treat patients aged 16 years or older in the adult ED (Table 1) [12,13,14,15,16,17]. This approach has also been adopted at our ED at the University Hospital in Bern.

There is little information on adolescent ED use in Switzerland. Data from other countries cannot readily be extrapolated, given the differences in age cut-offs and adolescent health services [3,18,19,20,21,22].

The overall goal of this study is to provide an overview of the presentations of patients aged 16–18 years to our tertiary ED, with a main focus on principle complaints and comparison to adults. The secondary objective is to formulate proposals for process optimisation, as these patients are a vulnerable and underserved population in Switzerland and beyond.

## 2. Materials and Methods

### 2.1. Study Design

We conducted a cross-sectional study of clinical presentations of adolescent patients aged 16 to 18 years presenting to the Inselspital adult ED in Bern. We analysed demographic and health-related data from January 2013 to December 2017. As this study used anonymised data, we obtained exemption from the cantonal ethics committee in Bern (No. Req-2020-01388).

### 2.2. Data Collection and Extraction

Data were extracted from the ED electronic medical record system (E.care BVBA, ED 2.1.3.0, Turnhout, Belgium) and included: demographics, triage category, mode of arrival, type of ED admission and discharge, and the principal complaint. We grouped principal complaints according to the resource consumption score of an emergency department consultation [4].

We analysed data from the electronic medical record system. The Swiss Emergency Triage Scale, implemented in our ED, uses five levels of urgency: (1) life-threatening situations (life/limb-threatening situations—immediate treatment); (2) highly urgent situations (potentially life-threatening situations—assessment and treatment within 20 min); (3) urgent situations (assessment and treatment within 120 min); (4) semi-urgent and (5) non-urgent situations.

### 2.3. Statistical Analysis

For our primary analysis, we formed two groups; adolescents aged 16–18 years and adults aged ≥18 years. Next, we further divided the group of ≥18-year-old patients into groups of ≥18–25 years, ≥25–35 years, and ≥35 years and compared these to 16–18 year old patients.

Descriptive summary statistics were computed as appropriate. Categorical data between groups were compared using chi-square tests. A linear regression was used to model the trend of the annual number of patients. The slope of the trend was presented with 95% confidence intervals (CI). The graphical comparison of the principal complaints included 95% confidence intervals of the proportions. We employed STATA^®^ 16.1 (StataCorp, The College Station, TX, USA) and Microsoft^®^ Excel^®^ for Microsoft 365 MSO (16.0.13929.20360, Office 2019, Redmond, WA, USA) for the analysis.

The WHO defines ‘adolescents’ as individuals in the 10–19 years of age group and ‘youths’ as the 15–24 years of age group, while ‘young people’ covers the age range of 10–24 years [9]. Therefore, we performed an additional sensitivity analysis of different subgroups within the group of patients older than 18 years. We formed the subgroups as follows: (1) ≥16–18 years, (2) ≥18–25 years, (3) ≥25–35 years, and (4) ≥35 years.

## 3. Results

### 3.1. Study Population

Our study included a total of 203,817 patient presentations to the adult ED in Bern between January 2013 and December 2017, among which 4930 (3.9%) were adolescents and 198,887 were adults (Figure 1). Among adolescents, there was a majority of female patients with 51.6% (*n* = 2543), but only 43.7% of adult patients were female (*n* = 86,812) (*p* < 0.001, Table 2). The median age for adolescents was 17 years (IQR 16–17) and 48 years for adults (IQR 32–66) (*p* < 0.001, Table 2).

Data from the descriptive analysis are summarised in Table 2.

### 3.2. Annual Changes

There was a significant association between the two study groups and the year of presentation (*p* < 0.001). The number of ED visits increased in both groups between 2013 and 2017. The percentage increase between 2013 and 2017 was different between the two groups, with a 5.9% increase in ED visits among adolescents compared to 25.8% among adults. The trend line shows a significant annual increase in the number of adult consultations of +2368 (95% CI: 1695, 3041, *p* = 0.002), compared with +26 (95% CI: −63, 115, *p* = 0.420) for adolescents (Table 2, Figure 2).

### 3.3. Presentation Characteristics

We found a significant difference between the two groups with respect to admission during the day (from 7 a.m. to 7 p.m.)—3001 adolescents (60.9%) and 136,217 adults (68.5%), *p* < 0.001)—and correspondingly, for admission during the night—from 7 p.m. to 7 a.m. (1929 adolescents (39.1%) and 62,670 adults (31.5%)). No significant association could be demonstrated between the two groups and admission during weekends (Saturday and Sunday) (*p* = 0.074), specific weekdays (*p* = 0.331), and admission on public holidays (*p* = 0.891, Table 2).

The mode of arrival differed between adolescents and adults: a greater proportion of adolescents were walk-in patients (*n* = 2674, 54.2%) when compared to adults (*n* = 89,254, 44.9%), while ambulance transfers were less common (10.3% for adolescents (*n* = 506) versus 14.7% for adults (*n* = 29,224), *p* < 0.001, Table 2).

With respect to the Swiss Emergency Triage Scale, we found a highly significant association between the two groups and the triage groups (life-threatening: adolescents 148 (3%) vs. adults 15,564 (7.8%), *p* < 0.001). The group of patients with urgent triages was greater in the adolescent group—with 65.3% (*n* = 3217) of patients—in comparison to the adult group, with 58.2% (*n* = 115,820). A total of 226 (4.6%) adolescents and 11,312 (5.7%) adults were treated in our ED resuscitation room (*p* = 0.001, Table 2).

### 3.4. Principle Complaint Groups

The three most common reasons for ED presentation in the adolescent group were (1) trauma (*n* = 1200, 24.3%); (2) musculoskeletal problems (including rheumatologic problems) (*n* = 677, 13.7%); and (3) psychiatric problems, including self-harm (*n* = 466, 9.5%). In the adult group, the most common reasons for ED presentation were (1) trauma (*n* = 28,685, 14.4%); (2) musculoskeletal problems (including rheumatologic problems) (*n* = 26,818, 13.5%); and (3) neurological problems (*n* = 23,108, 11.6%). Psychiatric problems among adults were less common, with 6.2% (Table 2, Figure 3).

### 3.5. Discharge

Most patients could be discharged, corresponding to 3490 adolescents (70.8%) and 103,880 adults (52.2%), while 562 (11.4%) of adolescent patients and 55,805 (28.1%) of adult patients were admitted as inpatients. Three adolescents had to be resuscitated in our ER. No adolescent patient died in our ED, while 332 (0.2%) adults died (Table 2).

### 3.6. Sensitivity Analysis

Detailed comparisons of the four age groups for sensitivity analysis ((1) ≥16–18 years, (2) ≥18–25 years, (3) ≥25–35 years, and (4) ≥35 years) are shown in Appendix A.

We found a significant difference between the first, second, and third groups with respect to gender, as there were more female patients in the first group ((1) 51.6% vs. (2) 47.1% vs. (3) 45.8%, *p* < 0.001). Older patients of groups two and three were more likely to be walk-in patients ((2) 55.9%, (3) 56%, (1) 54.2%, *p* < 0.001).

Significant associations between the different groups and their principle complaints were found, with more trauma cases among 16–18 year olds (Figure 4). No significant differences could be demonstrated with respect to the other categories (time of presentation, triage, and discharge).

## 4. Discussion

### 4.1. Summary of Findings

Adolescents between 16–18 years made up 2.5% (4930/198,887) of consultations at our tertiary ED between 2013 and 2017. Compared to adults, consultations did not significantly increase during the study period. In the adolescent age group, a higher proportion of patients were male. Adolescents were more likely to present during the night-time or as walk-in patients. They received lower triage scores and were more likely to be discharged. Adolescents presented most frequently with trauma, musculoskeletal symptoms, and psychiatric complaints. Trauma and psychiatric complaints were more frequent in adolescents than in adults. Traumatic complaints were more common for adolescents aged 16–18 years than in young adults (18–25 years) and adults (25–35, >35 years). Psychiatric complaints remained common up to age 35 but were significantly rarer in older patients.

### 4.2. Comparison with Other Studies

A UK study of 2014 found that male patients presented more frequently in all age groups between 13 and 17 years [7]. Male adults presented more often to our ED than female adults (male 56.4%, female 43.6%).

There are several studies that have examined the emergency perceptions of adults and children. For this purpose, a study from the United States evaluated statistical data from the ED in 2011. The authors found that adults in the outpatient setting presented most frequently with sprains and strains, superficial trauma, abdominal pain, and back pain. The most common reasons for hospitalisation were sepsis, pneumonia, heart failure, and COPD [3]. A 2015 cross-sectional study found that pneumonia, heart failure, nonspecific chest pain, and sepsis mostly led to hospitalisation in adults [23].

Among children ages 1–17, trauma, otitis media, and febrile illness were the most common diagnoses in the outpatient setting. Children were most often hospitalised with acute bronchitis, pneumonia, asthma, and appendicitis [3]. Similarly, a study by Montalbano et al. found that children presented most frequently with upper respiratory tract infections, fever, and otitis media [19].

To the best of our knowledge, only a few studies have examined the reasons that adolescents between the ages of 16–18 years present to the ED. A study in the UK published in 2014 found that teenagers between the ages of 13–17 years most commonly present with injuries, abdominal pain, or self-harm [7].

According to our data, ED presentation among adolescents aged 16–18 differs from presentation among children aged 1–17 and adults older than 18 years [3,19,24].

The Swiss Medical Association for Paediatrics states that children should not be viewed as ‘small adults’ [25]. However, adolescents who are treated in adult EDs from the age of 16 are at risk of receiving diagnostic testing and therapy adapted to adults. Multiple international studies have found that ED visits for psychiatric purposes are rising in young patients [26,27,28], and this is further exacerbated by the COVID-19 pandemic [29]. However, a study in the United States found that only 16% of all patients are seen by a mental health professional [26]. Another study in Australia found that children with mental health problems presenting to the ED receive care after a significantly longer delay than patients presenting with physical problems [30]. Our tertiary ED does not work with specific resources adapted to non-adult patients, for example, paediatric specialists as children psychiatrists or traumatologists, which could exacerbate these circumstances.

Our data search showed that only three adolescent patients aged 16–18 needed to be resuscitated; none of these patients died in our ER. According to a study from our department, it can be estimated that approximately 14 adult patients (over 18 years of age) die under or after resuscitation while still in our emergency ward per month [31]. However, our data only reflect the ER stay and do not include the mortality rate of patients who die during hospitalisation after resuscitation.

### 4.3. Strengths and Limitations

For this study, only data collected during patient triaging were used. Accordingly, the recorded symptoms were analysed on the basis of the information provided at admission. More detailed studies should be conducted to analyse specific groups.

One significant limitation was that we could only conduct a cross-sectional study with the available data.

For further in-depth analysis, more detailed data should be evaluated, including the discharge diagnosis. As the Inselspital ED is a large, university ED, there might be a selection bias towards more severe diagnoses. In the city of Bern alone, there are seven other, smaller emergency wards (Hirslanden Klinik (Salemspital, Klinik Beausite, Klinik Permanence), Tiefenauspital Inselgruppe, and Lindenhofgruppe (Lindenhofspital, Klinik Sonnenhof, City Notfall)). According to our data search, these clinics have not published data about adolescent emergency presentation. Data from other EDs should also be analysed to obtain a more comprehensive picture of the adolescent population.

### 4.4. Relevance/Discussion of the Findings

In many large hospitals in Switzerland, adolescents from 16 years and older are treated in adult EDs, but little has been published on this specific patient population. This study aimed to provide an overview of emergency presentations in patients between 16–18 years of age to our tertiary ED.

The main purpose of this study was to provide an overview of adolescent presentation compared with adults in an adult ED. Since many large hospitals in Switzerland treat adolescents over 16 years of age in an adult ED, this topic has national relevance. There is little literature determining why a limit at 16 years might be useful. One consideration is that children with chronic conditions may make a planned transition between paediatric and adult medicine at age 16 and thus are confronted with adult emergency rooms at an earlier age [10]. To guarantee quality-assured care for these patients, we believe that certain areas require special attention in order to improve the processes of care for adolescent patients. As an American publication has already suggested for children, adolescents should be treated with specific procedures in the emergency ward [32].

Adolescent patients should not be viewed as small adults and, consequently, should not be treated as such. Younger patients need more time to be fully informed about the diagnosis and therapy of their condition [25]. Accordingly, consultation of paediatric specialists might be considered before initiating appropriate treatment in adolescents. Patients with psychiatric problems should be carefully triaged, and specialists should be consulted at an early stage [2]. Our emergency centre is continuously committed to optimising processes based on resource analysis of specific patient populations [4,33,34,35]. This should continue in relation to the particularly vulnerable population of adolescent patients. Therefore, we suggest an established process flow for standardised care in adolescent patients, especially in patients with trauma and psychiatric problems.

## 5. Conclusions

This study provides evidence that the principle complaints in adolescent patients under 18 years of age significantly differ from those in older patients. Age cut-offs for transition of care may not reflect the healthcare needs of this population. Further studies are needed to determine the reasons that adolescents and young adults seek ED care. Coupled with resource analysis, these studies could help to improve care processes for this vulnerable population.

## Figures and Tables

**Figure 1 ijerph-18-09578-f001:**
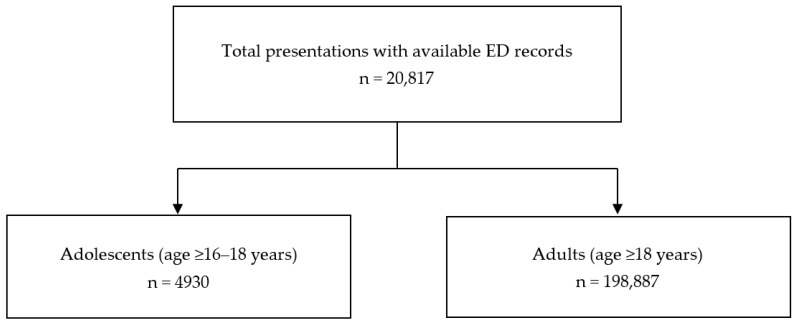
Flow chart of study population.

**Figure 2 ijerph-18-09578-f002:**
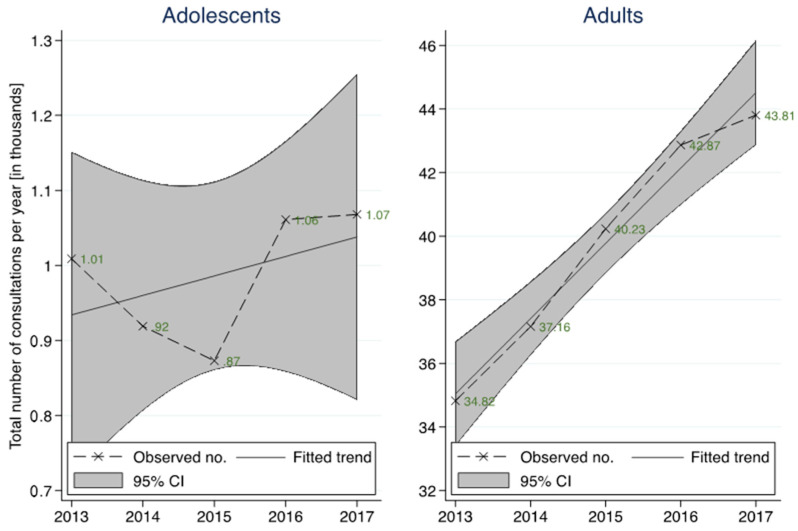
Observed case numbers with a linear trend including 95% CI of the trend for adolescents (16–18) and adults from 2013–2017.

**Figure 3 ijerph-18-09578-f003:**
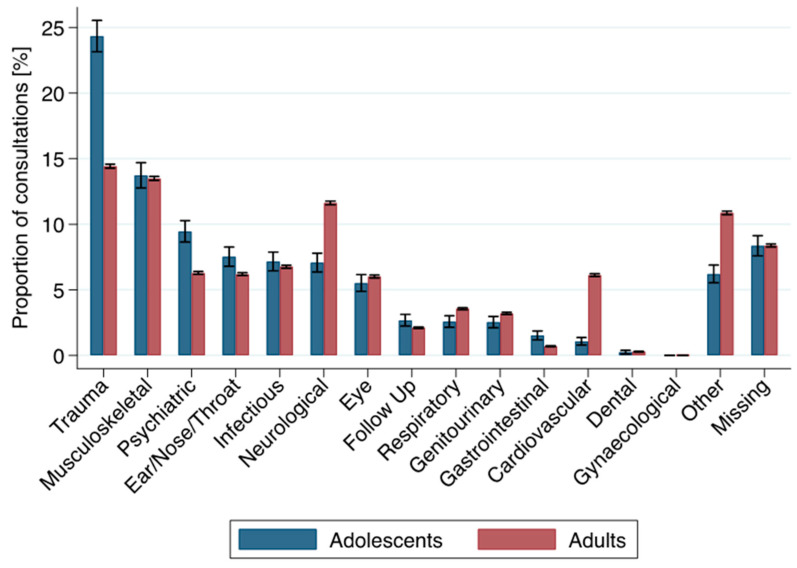
Proportion—with 95% CI—of principle complaint groups among adolescents and adults, sorted by frequency in the adolescents group.

**Figure 4 ijerph-18-09578-f004:**
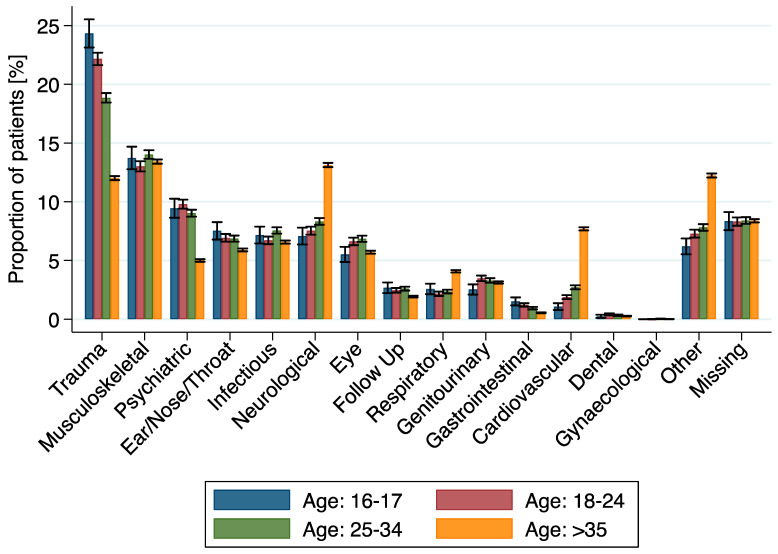
Proportion of principle complaint groups among different age subgroups, sorted by frequency in the age group 16–17.

**Table 1 ijerph-18-09578-t001:** Emergency room age limits of major hospitals in Switzerland.

Hospital	Paediatric ED Age Limit (years)
Aarau	18
Basel University	18
Geneva	16 ^1^
Lucerne	16 ^2^
Winterthur	16
Zurich University	16

^1^: 18 years for patients with psychiatric complaints. ^2^: 20 years for patients with underlying chronic conditions. ED = emergency department.

**Table 2 ijerph-18-09578-t002:** Demographics, type of ED use, and chief complaints of adolescents aged 16–18 years compared to adults ≥ 18 years.

Category	Adolescents 16–18 (*n* = 4930)		Adults ≥18 (*n* = 198,887)		*p*
**Year, *n* (%)**					<0.001
2013	1009	(20.5)	34,821	(17.5)	
2014	919	(18.6)	37,159	(18.7)	
2015	873	(17.7)	40,234	(20.2)	
2016	1061	(21.5)	42,867	(21.6)	
2017	1068	(21.7)	43,806	(22.0)	
**Presentation Date and Time**					
Saturday or Sunday admission (00:00–23:59), *n* (%)	1497	(30.4)	58,061	(29.2)	0.074
Public and cantonal (Bern) holidays, *n* (%)	98	(2.0)	4009	(2.0)	0.891
Night-time admissions (19:00–06:59), *n* (%)	1929	(39.1)	62,670	(31.5)	<0.001
**Day of the week, *n* (%)**					0.331
Monday	712	(14.4)	29,725	(14.9)	
Tuesday	666	(13.5)	27,113	(13.6)	
Wednesday	675	(13.7)	27,296	(13.7)	
Thursday	663	(13.4)	27,524	(13.8)	
Friday	717	(14.5)	29,168	(14.7)	
Saturday	737	(14.9)	29,912	(15.0)	
Sunday	760	(15.4)	28,149	(14.2)	
**Type of Admission, *n* (%)**					<0.001
Ambulance	506	(10.3)	29,224	(14.7)	
General practitioner	170	(3.4)	10,969	(5.5)	
External hospital	243	(4.9)	15,231	(7.7)	
Police	73	(1.5)	2154	(1.1)	
Air rescue	65	(1.3)	2465	(1.2)	
Repatriation	5	(0.1)	319	(0.2)	
Walk-in	2674	(54.2)	89,254	(44.9)	
Internal referral	180	(3.7)	8238	(4.1)	
Urgent care centre/doctor	49	(1.0)	1787	(0.9)	
Other	12	(0.2)	769	(0.4)	
Missing information	953	(19.3)	38,477	(19.3)	
**Triage, *n* (%)**					<0.001
Life-threatening	148	(3.0)	15,564	(7.8)	
Highly urgent	889	(18.0)	46,193	(23.2)	
Urgent	3217	(65.3)	115,820	(58.2)	
Semi-urgent	448	(9.1)	13,746	(6.9)	
Non-urgent	125	(2.5)	3209	(1.6)	
Missing Information	103	(2.1)	4355	(2.2)	
**Resuscitation room treatment, *n* (%)**					0.001
No	4704	(95.4)	187,573	(94.3)	
Yes	226	(4.6)	11,312	(5.7)	
**Principle complaint, *n* (%)**					<0.001
Psychiatric problem (including self-harm)	466	(9.5)	12,494	(6.3)	
Musculoskeletal + rheumatologic problems	677	(13.7)	26,818	(13.5)	
Gastrointestinal problems	75	(1.5)	1389	(0.7)	
Respiratory problems	127	(2.6)	7061	(3.6)	
Neurological problems	349	(7.1)	23,108	(11.6)	
Cardiovascular problems	53	(1.1)	12,174	(6.1)	
Infectious disease, including skin problems	353	(7.2)	13,447	(6.8)	
Obstetric or gynaecological problems	0	(0.0)	26	(0.0)	
Dental problems	12	(0.2)	580	(0.3)	
Eye problems	272	(5.5)	11,963	(6.0)	
Other	306	(6.2)	21,610	(10.9)	
Trauma	1200	(24.3)	28,685	(14.4)	
Genitourinary problems	125	(2.5)	6359	(3.2)	
Ear/nose/throat problems	371	(7.5)	12,311	(6.2)	
Follow-up	132	(2.7)	4205	(2.1)	
Missing information	412	(8.4)	16,657	(8.4)	
**Discharge, *n* (%)**					<0.001
Death	0	(0.0)	332	(0.2)	
Discharge home	3490	(70.8)	103,880	(52.2)	
Hospital admission	562	(11.4)	55,805	(28.1)	
Transfer to external hospital	320	(6.5)	17,426	(8.8)	
Other	44	(0.9)	1711	(0.9)	
Not specified	514	(10.4)	19,724	(9.9)	
Missing information	0	(0.0)	9	(0.0)	

## Data Availability

Not Applicable.

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
