# Peer review of "Clinical Presentations of Adolescents Aged 16–18 Years in the Adult Emergency Department"

_ijerph, 2021, doi:10.3390/ijerph18189578_

Round 1
Reviewer 1 Report
Dear Authors,
- In my opinion it would be better to write more about the University Hospital especially about missing departments (if applicable) and about other ED in the city (e.g.: how many?)
- The references should be change according rules of the IJERPH
- The text needs some minor spell check
- You are using 5 level of urgency (The Swiss Emergency Triage Scale) - and in some papers you can read about 4 level of urgency https://www.ncbi.nlm.nih.gov/pmc/articles/PMC6039392/ - can you describe the difference
Author Response
Review 1
- In my opinion it would be better to write more about the University Hospital especially about missing departments (if applicable) and about other ED in the city (e.g.: how many?)
Author Response:
Thank you for your input. We only conducted a retrospective study with data from our ED. Unfortunately, no other department’s data could be used for this analysis. We added the number of EDs in the city of Bern to the study, these have not published any data regarding adolescents.
- The references should be change according rules of the IJERPH
Author Response:
Thank you for this comment, the referencing style has been adapted accordingly.
- The text needs some minor spell check
Author Response:
Another spell check has been conducted before submitting the final version.
- You are using 5 level of urgency (The Swiss Emergency Triage Scale) - and in some papers you can read about 4 level of urgency https://www.ncbi.nlm.nih.gov/pmc/articles/PMC6039392/ - can you describe the difference
Author Response:
A new, revised version of the SETS now only uses 4 levels of urgency (revision of 2018). Since the data come from the years 2013-2017, our patients were still divided into 5 groups, which is why the older system was used in this study.
Reviewer 2 Report
thank you for the opportunity to read the manuscript. The authors raise an important topic of medical interventions in general ED for pediatric patients. Before publication, please the authors to make the following corrections:
1) in Figure 2, it is worth adding a description of the dashed line to the legend, and also adding numerical values to the results in individual years.
2) Figure 3 does not bring significant value to the work. Maybe it is worth ordering the data in descending order to show the trend? The lack of order can also be seen in Figure 4 (maybe it is worth at least to sort the cases alphabetically?)
3) Can the authors supplement the data with cardiac arrest cases among adolescents and adults? No deaths have been reported among adolescents, but there may have been cases of CPR implemented. This result may further enable a comparative scale between adult ED and pediatric SED.
4) In tables and supplements, P values are shown in bold. I propose to highlight ONLY statistically significant P values.
5) References should be completed. Out of the 26th position, only 7 meet the criteria of new (not older than 3 years). In addition, many of the items refer to websites rather than scientific publications. Please post some recent research related to the topic, here are some examples:
a) Remick K, Gausche-Hill M, Joseph MM, Brown K, Snow SK, Wright JL, et al. Pediatric readiness in the emergency department. Pediatrics, 2018; 142 (5): e20182459.
(current ED actions towards pediatric patients)
b) Sosnowska-Mlak O, Curt N, Pinet-Peralta LM. Survival in sudden cardiac arrest in emergency room: case-control study. Crit. Care Innov. 2019; 2 (3): 1-10.
(life-threatening conditions in ED - causes and procedures)
c) Sholokhova D, Sviatoslav M. The pediatric trauma patient profile from the perspective of the emergency medical service. Crit. Care Innov. 2019; 2 (2): 1-8.
(profile of pediatric trauma patients during emergency interventions)
Author Response
Review 2
- In Figure 2, it is worth adding a description of the dashed line to the legend, and also adding numerical values to the results in individual years.
Author Response:
As suggested, we included the numerical values to the results and altered the figure description.
- Figure 3 does not bring significant value to the work. Maybe it is worth ordering the data in descending order to show the trend? The lack of order can also be seen in Figure 4 (maybe it is worth at least to sort the cases alphabetically?)
Author Response:
The graphs did indeed lack some order. We now sorted both figures by frequency in the adolescent group (16-18 years / 16-17 years) to provide more clarity and adjusted the figure description accordingly.
- Can the authors supplement the data with cardiac arrest cases among adolescents and adults? No deaths have been reported among adolescents, but there may have been cases of CPR implemented. This result may further enable a comparative scale between adult ED and pediatric SED.
Author Response:
Thank you for your input. This additional information has been added to the results and to the discussion.
“Our data search showed that only three adolescent patients aged 16-18 needed to be resuscitated, none of these patients died in our ER. According to a study from our department, it can be estimated that per month about 14 adult patients (over 18 years of age) die under or after resuscitation while still in our emergency ward [32]. However, our data only reflects the ER stay and does not include the mortality rate of patients who die during hospitalisation after resuscitation”.
- In tables and supplements, P values are shown in bold. I propose to highlight ONLY statistically significant P values.
Author Response:
All P values have been checked and only significant values are written in bold.
- References should be completed. Out of the 26th position, only 7 meet the criteria of new (not older than 3 years). In addition, many of the items refer to websites rather than scientific publications. Please post some recent research related to the topic, here are some examples:
- a) Remick K, Gausche-Hill M, Joseph MM, Brown K, Snow SK, Wright JL, et al. Pediatric readiness in the emergency department. Pediatrics, 2018; 142 (5): e20182459.
(current ED actions towards pediatric patients)
- b) Sosnowska-Mlak O, Curt N, Pinet-Peralta LM. Survival in sudden cardiac arrest in emergency room: case-control study. Crit. Care Innov. 2019; 2 (3): 1-10.
(life-threatening conditions in ED - causes and procedures)
- c) Sholokhova D, Sviatoslav M. The pediatric trauma patient profile from the perspective of the emergency medical service. Crit. Care Innov. 2019; 2 (2): 1-8.
(profile of pediatric trauma patients during emergency interventions)
Author Response:
This was a very helpful point in reviewing our paper, thank you. We conducted another data research and found some papers that could be included in addition that meet the reference criteria.
We certainly hope that our paper is soon to be accepted and published in the International Journal of Environmental Research and Public Health.